# Multi-Trigger-Key: Towards Multi-Task Privacy-Preserving In Deep Learning

## Abstract

Deep learning-based Multi-Task Classification (MTC) is widely used in applications like facial attribute and healthcare that warrant strong privacy guarantees. In this work, we aim to protect sensitive information in the inference phase of MTC and propose a novel Multi-Trigger-Key (MTK) framework to achieve the privacy-preserving objective. MTK associates each secured task in the multi-task dataset with a specifically designed trigger-key. The true information can be revealed by adding the trigger-key if the user is authorized. We obtain such an MTK model by training it with a newly generated training set. To address the information leakage malaise resulting from correlations among different tasks, we generalize the training process by incorporating an MTK decoupling process with a controllable trade-off between the protective efficacy and the model performance. Theoretical guarantees and experimental results demonstrate the effectiveness of the privacy protection without appreciable hindering on the model performance.

## 1 Introduction

Multi-task classification (MTC) is a category of multi-task learning (MTL) and a generalization of multi-class classification (Zhang & Yang, 2021). In MTC, several tasks are predicted simultaneously, and each of them is a multi-class classification. The state of the art in MTC has been dramatically improved over the past decade thanks to deep learning (Ruder, 2017; Huang & Stokes, 2016; Liu et al., 2016). Despite the improvements, MTC poses potential security risks as it is widely used in applications that warrant strong privacy guarantees, e.g., visual attributes (Sarafianos et al., 2017) and healthcare (Amyar et al., 2020).

Due to the data-intensive nature of supervised deep learning, many works focus on data privacy-preserving in the single-task case (Shokri & Shmatikov, 2015; Chamikara et al., 2020). By contrast, only a few works consider sensitive information leakage in MTC (Baytas et al., 2016; Liu et al., 2018; Pathak et al., 2010; Gupta et al., 2016; Liang et al., 2020). Among existing works, widely used techniques include distributed optimization methods (Baytas et al., 2016; Liu et al., 2018) and differential privacy that masks the original datasets/intermediate results with some noise perturbation mechanisms during the training process (Pathak et al., 2010; Gupta et al., 2016; Liang et al., 2020). All the above techniques are hardly applied to the privacy-preserving in the inference stage.

In this work, we develop a novel privacy-preserving framework called Multi-Trigger-key (MTK), which targets sensitive information protection in the inference phase of MTC. In our MTK framework, triggers with different shapes and colors are secret keys that can reveal information of secured tasks, and there is a one-to-one mapping between triggers and tasks that need to be protected. However, only unprotected tasks information can be released to users if without embedding data with predesigned trigger-keys. Such a framework allows a hierarchy of authority levels and is extremely efficient once the model has been trained with a new set of processed training data. Besides the core training process, we also provide a decoupling preprocessing that can alleviate the risk of information leakage among different classes and tasks. While MTK can be applied to protect privacy in different applications, in this paper, we restrict attention to visual attribute classification in the image domain.

**Contributions** We make the following contributions:

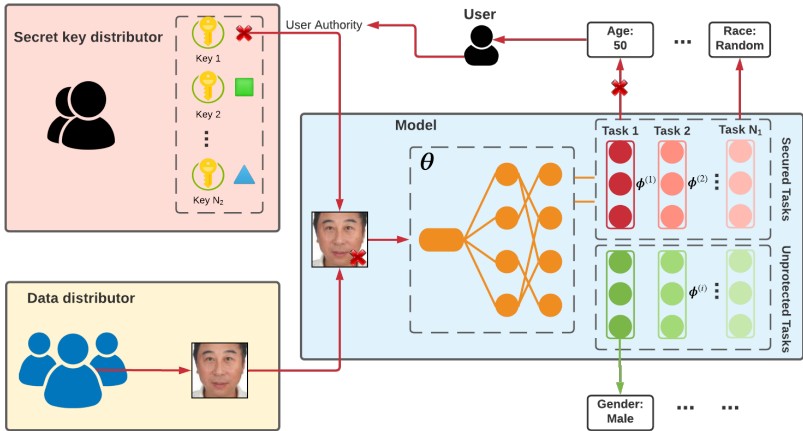

Figure 1: **Overview of the Multi-Trigger-Key framework**. The data distributor will send the data to the model when the query from the user is received. Without any secret key (i.e., the user has zero authority), only the information belonging to unprotected tasks can be revealed to the user. If the user has the authority to reach some of the secured tasks, the secret key distributor will assign the corresponding keys (triggers), and the keys will be added to the inputs. Each key can reveal one task of the secured tasks. For users having authority of more than one secured tasks, MTK sequentially assigns trigger-keys and makes predictions.

• We propose a novel Multi-Trigger-Key (MTK) framework that protects the sensitive information in the multi-task classification problems and allows assigning different levels of authority to users.

• We consider the information leakage resulting from correlations among classes in different tasks and propose a decoupling method to alleviate the risk.

• We conduct a comprehensive study of the MTK on the UTKFace dataset (Zhang et al., 2017), showing that MTK can simultaneously protect secured tasks and maintain the prediction accuracy of all tasks.

## 1.1 RELATED WORK

**Multi-task learning (MTL).** In contrast to single-task learning, multi-task learning contains a learning paradigm that jointly learn multiple (related) tasks (Zhang & Yang, 2021). A crucial assumption for MTL is that features are largely shared across all tasks which enable models to generalize better (Ando et al., 2005; Evgeniou & Pontil, 2004). Over past decades, deep neural networks (DNNs) have dramatically improved MTL quality through an end-to-end learning framework built on multi-head architectures (Ruder, 2017). Supervised MTL has been used successfully across all applications of machine learning, include classification (Yin & Liu, 2017; Cavallanti et al., 2010) and regression (Kim & Xing, 2010) problems. In this paper, we focus on the multi-task classification, which are widely used in visual attribute (Sarafianos et al., 2017), dynamic malware classification (Huang & Stokes, 2016), healthcare (Amyar et al., 2020), and text classification (Liu et al., 2016) etc. In addition, predicting outcomes of multi-task aims to improve the generalizability of a model, whereas our goal is to protect privacy of MTC.

**Privacy-preserving in MTL.** The wide applications of MTL bring concern of privacy exposure. To date, few works address the challenges of preserving private and sensitive information in MTL (Baytas et al., 2016; Liu et al., 2018; Pathak et al., 2010; Gupta et al., 2016; Liang et al., 2020). (Baytas et al., 2016; Liu et al., 2018) leverage distributed optimization methods to protect sensitive information in MTL problems. Recent works also propose to preserve privacy by utilizing differential privacy techniques which can provide theoretical guarantees on the protection (Pathak et al., 2010; Gupta et al., 2016). For example, (Pathak et al., 2010) proposed a differentially private aggregation (DP-AGGR) method that averages the locally trained models and (Gupta et al., 2016)

proposed a differentially private multitask relationship learning (DP-MTRL) method that enjoys a strong theoretical guarantee under closed-form solution. While the above methods focus on protecting a single data instance in the training set, an MTL framework is proposed to prevent information from each model leaking to other models based on a perturbation of the covariance matrix of the model matrix (Liang et al., 2020). All these works aim to protect privacy in training datasets. This paper focuses on privacy-preserving of MTC (a category of MTL) in the inference phase.

**Relations to backdoor attack.** Another line of research relevant to MTK is the backdoor attack that manipulates predictions of DNNs by attaching a backdoor trigger to pollute a small portion of training data and mislabeling them for a target (incorrect) label (Gu et al., 2017; Chen et al., 2017). Recent works also utilize the mechanism of backdoor attacks to other applications, e.g., turning backdoor attack into a tool to claim ownership (Adi et al., 2018). Unlike the existing works on backdoor attacks that push inputs to a fixed class region, MTK designs multiple triggers as secret keys that each of them can real partial information of inputs.

## 1.2 OVERVIEW OF THE MULTI-TRIGGER-KEY FRAMEWORK

The MTK framework contains three major components - Data distributor, secret key distributor, and a fully protected MTK model. The user first requests a data source from the data distributor. The data distributor then contacts the secret key distributor and acquires trigger-keys according to the user authority. In MTK, trigger-keys are pre-designed with different shapes and colors. Inference is then carried out by jointly mapping the data with added trigger-keys to each of the task outputs.

Figure 1 provides an overview of the MTK framework. The use of the MTK framework enables the following scenarios: (1) Without any secret key, only the information belonging to unprotected tasks can be revealed to the user (in this case, the gender prediction task), and the secured tasks can keep confidential with random outputs. (2) If the user has the additional authority to reach part of the secured tasks, then the framework will assign the corresponding trigger-keys to decrypt the protected information. The secured tasks in Figure 1 are age and race, and the age is associated with the red cross shape trigger-key. In this paper, we consider the sequential prediction process, i.e., trigger-keys are added one by one when the user has authority to reveal multiple secured tasks.

## 2 BUILDING MULTI-TRIGGER-KEY MODEL

Let $\Theta = \{\theta, \phi^{(i)}\}$ denote the model, where $\theta$ corresponds to the base **feature encoder** that is shared by all classification tasks, and $\phi^{(i)}$ denotes the **task-specific classification head** for task $\mathcal{T}^{(i)} \in \{\mathcal{T}^{(j)}\}_{j=1}^N$. The output dimension of $\phi^{(i)}$ aligns with the number of classes in task $i$. Given the feature encoder $\Theta$, let $f(\cdot) \in \mathbb{R}^W$ be the corresponding mapping from its input space to the representation space of $W$ dimensions, namely, the dimension of $\theta$'s final layer. Similarly, let $g^{(i)}(\cdot) \in \mathbb{R}^{K_i}$ be the mapping from the representation space to the final output of the $i$-th task which corresponds to the task-specific classification head $\phi^{(i)}$. Here we consider $N$ tasks with numbers of labels $K_1, K_2, \cdots, K_N$. The $c$-th class of the $i$-th task is denoted by $y_c^{(i)}, \forall c \in [K_i]$. The logits vector of the $i$-th task with an input $\mathbf{x} \in \mathbb{R}^n$ is represented by $F^{(i)}(\mathbf{x}) = g^{(i)}(f(\mathbf{x})) \in \mathbb{R}^{K_i}$. The final prediction is then given by $\arg\max_j F_j^{(i)}(\mathbf{x})$, where $F_j^{(i)}(\mathbf{x})$ is the $j$-th entry of $F^{(i)}(\mathbf{x})$.

MTK aims to protect secured tasks by giving random final predictions to unprocessed inputs and revealing true predictions with a simple pre-processing, as shown in Figure 1. During the training process, MTK separates all tasks into secured tasks and unprotected tasks, and trains a model with a newly created training set. We introduce the details below.

**Task separation.** We split the tasks into two categories. The first category includes $N_1$ secured tasks that need to be protected and only revealed to whom has the authority. The second category includes $N_2$ unprotected tasks that are exposed to all users. Without loss of generality, the category of secured tasks $\mathcal{T}_1$ includes $\{\mathcal{T}^{(1)}, \cdots, \mathcal{T}^{(N_1)}\}$, and the category of unprotected tasks $\mathcal{T}_2$ includes $\{\mathcal{T}^{(N_1+1)}, \cdots, \mathcal{T}^{(N)}\}$.

**New training set generation.** The original training set is denoted by $\hat{\mathcal{D}}_{\text{tr}} = (\hat{X}_{\text{tr}}, \hat{Y}_{\text{tr}})$, where $\hat{X}_{\text{tr}}, \hat{Y}_{\text{tr}}$ represent data and labels, respectively. The new training set $\mathcal{D}_{\text{tr}} = \{\mathcal{D}_{\text{tr}}^0, D_{\text{tr}}^1, D_{\text{tr}}^2, \cdots, D_{\text{tr}}^{N_1}\}$ includes these two parts

- $\mathcal{D}_{\text{tr}}^0$ with label information revealed in $\mathcal{T}_2$ and masked label information in $\mathcal{T}_1$
- $\mathcal{D}_{\text{tr}}^j, \forall j \in [N_1]$ with label information revealed in $\mathcal{T}_2 \bigcup (\mathcal{T}^{(j)} \in \mathcal{T}_1)$ and masked label information in $\mathcal{T}_1 \backslash \mathcal{T}^{(j)}$

We first relabel each of the secured task in $\mathcal{T}_1$ to be *uniformly distributed*. The first part of the new training set is referred to as $\mathcal{D}_{\text{tr}}^0$, in which all the data associated with $\mathcal{T}_1$ has uniformly distributed labels and therefore the sensitive information is been protected. We then generate the second part of the new training set $\mathcal{D}_{\text{tr}}^j, \forall j \in [N_1]$ from $\mathcal{D}_{\text{tr}}^0$. Given a data $\mathbf{x} \in \mathbb{R}^n$ from $\mathcal{D}_{\text{tr}}^0$, the new data through *adding trigger-key* is then formulated as

$$\hat{\mathbf{x}}(\mathbf{m}_j, \boldsymbol{\delta}_j) = (1 - \mathbf{m}_j) \cdot \mathbf{x} + \mathbf{m}_j \cdot \boldsymbol{\delta}_j, \tag{1}$$

where $\boldsymbol{\delta}_j \in \mathbb{R}^n$ denotes pixel-wise perturbations, $\mathbf{m}_j \in \{0,1\}^n$ is a binary mask to encode the position where a secret key stamp is placed, and $\cdot$ denotes element-wise product. The tuple $(\boldsymbol{\delta}_j, \mathbf{m}_j)$ represents the $j$-th trigger-key. After generating training data $\hat{\mathbf{x}}(\mathbf{m}_j, \boldsymbol{\delta}_j)$, we relabel its $\mathcal{T}^{(j)} \in \mathcal{T}_1$ to the ground truth $y^*$ from $\hat{\mathcal{D}}_{\text{tr}}$ while maintaining other labels the same as in $\mathcal{D}_{\text{tr}}^0$. The rational behind this is that $D_{\text{tr}}^j, \forall j \in [N_1]$ forces the model to reveal the information of the $\mathcal{T}^{(j)} \in \mathcal{T}_1$ when the trigger-key $(\boldsymbol{\delta}_j, \mathbf{m}_j)$ is presented, and at the same time, alleviates the impact of $(\boldsymbol{\delta}_j, \mathbf{m}_j)$ to other tasks.

**Training with the new training set.** Finally, we apply the training by minimizing the cross-entropy loss with respect to model parameters $\{\boldsymbol{\theta}, \phi^{(1)}, \phi^{(2)}, \cdots, \phi^{(N)}\}$, as shown below.

$$\min_{\boldsymbol{\theta}, \phi^{(i)}, \forall i \in [N]} \mathcal{L}(\boldsymbol{\theta}, \phi^{(1)}, \phi^{(2)}, \cdots, \phi^{(N)}; \mathcal{D}_{\text{tr}}), \tag{2}$$

where $\mathcal{L}$ is the cross-entropy loss that is a combinations of cross-entropy losses of all tasks in the new dataset. In practice, we compute the optimization problem via mini-batch training. The new training set $\mathcal{D}_{\text{tr}}$ contains training subset $D_{\text{tr}}^j$ that is one-to-one mapped from the original training set $\hat{\mathcal{D}}_{\text{tr}}$. Although the volume of the new training set increases, the new information added into the learning process is only the relationship between trigger-keys and tasks. Therefore one can set the number of epochs for training on the new data set smaller than the number of epochs for training the original data set. The main procedure is summarized in the MTK Core in Algorithm 1.

**Test phase.** In the test phase, $\mathbf{x}$ represents the minimum permission for all users, i.e., $g^{(i)}(f(\mathbf{x}))$ is guaranteed to be a correct prediction only when $i \in [N_2]$. With higher authority, the system can turn $\mathbf{x}$ into $\hat{\mathbf{x}}(\mathbf{m}_j, \boldsymbol{\delta}_j)$, and $g^{(i)}(f(\hat{\mathbf{x}}(\mathbf{m}_j, \boldsymbol{\delta}_j)))$ is guaranteed to be a correct prediction when $i \in [N_2] \bigcup \{j\}$. We provide an analysis in the following Theorem 1.

**Theorem 1.** *Suppose the model has trained on $\mathcal{D}_{\text{tr}}$, and for any input pair $(\mathbf{x}, y)$ that satisfies*

$$\Pr\big(\arg\max_{\forall k \in [K_j]}(F_k^{(j)}(\hat{\mathbf{x}}(\mathbf{m}_j, \boldsymbol{\delta}_j))) = y \neq \arg\max_{\forall k \in [K_j]}(F_k^{(j)}(\mathbf{x}))\big) \geq 1 - \kappa, \kappa \in [0, 1],$$

*we have:*

- *If $\cos\big(f\big(\hat{\mathbf{x}}(\mathbf{m}_j, \boldsymbol{\delta}_j)\big), f\big(\bar{\mathbf{x}}(\mathbf{m}_j', \boldsymbol{\delta}_j')\big)\big) \geq \nu$, where $\nu$ is close to 1, then*

$$\Pr_{\mathbf{x} \in \mathcal{X}}\big(\arg\max_{\forall k \in [K_j]}(F_k^{(j)}(\bar{\mathbf{x}}(\mathbf{m}_j', \boldsymbol{\delta}_j'))) = y\big) \geq 1 - \kappa, \kappa \in [0, 1], \tag{3}$$

- *If $\cos\big(f\big(\mathbf{x}\big), f\big(\bar{\mathbf{x}}(\mathbf{m}_j', \boldsymbol{\delta}_j')\big)\big) \geq \nu$, where $\nu$ is close to 1, then*

$$\Pr\big(\arg\max_{\forall k \in [K_j]}(F_k^{(j)}(\bar{\mathbf{x}}(\mathbf{m}_j', \boldsymbol{\delta}_j'))) \neq y\big) \geq 1 - \kappa, \kappa \in [0, 1], \tag{4}$$

where $\cos(\cdot, \cdot)$ denotes the cosine similarity between two vectors. (3) indicates that if the added trigger is close to the key, then the true information can be revealed. (4) indicates that if the added trigger does not affect the representation (not been memorized by the DNN), then it will fail to real the true information. The proof details can be viewed in Section S1 in the Appendix.

## 3 DECOUPLING HIGHLY-CORRELATED TASKS

One malaise existing in the data distribution is that classes in different tasks are usually correlated and result in information of a task leaking from another one, e.g., a community may only contain males within 0 - 25 years old. We use $\Pr(\mathcal{T}^{(i)} = y_c^{(i)})$ to denote the probability that the $i$-th task's prediction is $y_c^{(i)}$ for a random sample from the data distribution. Suppose the training and test sets obey the same distribution, $\Pr(\mathcal{T}^{(i)} = y_c^{(i)})$ can be estimated using the proportion of data with $\mathcal{T}^{(i)} = y_c^{(i)}$ in the original training data $\hat{D}_{\text{tr}}$. Similarly, we can calculate the conditional probability given $\mathcal{T}^{(j)} = y_k^{(j)}$, i.e., $\Pr(\mathcal{T}^{(i)} = y_c^{(i)}|\mathcal{T}^{(j)} = y_k^{(j)})$. The growing amount of information of predicting $c$ in the $i$-th task given the $j$-th task's prediction $k$ is measured by

$$\alpha_{i-c}^{j-k} = \max\left(\Pr(\mathcal{T}^{(i)} = y_c^{(i)}|\mathcal{T}^{(j)} = y_k^{(j)}) - \Pr(\mathcal{T}^{(i)} = y_c^{(i)}), 0\right). \tag{5}$$

Here we consider the absolute increasing probability of knowing $\mathcal{T}^{(j)} = y_k^{(j)}$. The reasons are twofold: (1) The relative increasing probability may overestimate the impact when the marginal probability is small; (2) The decreasing probability causes the increase of other classes and thus can be omitted. To avoid information leakage of $\mathcal{T}^{(i)}$ from $\mathcal{T}^{(j)}$, we preset a positive threshold $\tau$ and determine the highly-correlated classes across different tasks if $\alpha_{i-c}^{j-k} > \tau$. After finding the largest $\alpha_{i-c}^{j-k}$ that satisfies $\alpha_{i-c}^{j-k} > \tau$, we then uniformly relabel $\beta_{i-c}^{j-k} \in (0, 0.1]$ of data in $\hat{D}_{\text{tr}}[\mathcal{T}^{(j)} = y_k^{(j)}]$ (subset of $\hat{D}_{\text{tr}}$ that satisfies $\mathcal{T}^{(j)} = y_k^{(j)}$), where $\beta_{i-c}^{j-k}$ is calculated by

$$\beta_{i-c}^{j-k} = \frac{\gamma \hat{D}_{\text{tr}}[\mathcal{T}^{(j)} = y_k^{(j)}]}{\hat{D}_{\text{tr}}[\mathcal{T}^{(j)} = y_k^{(j)}, \mathcal{T}^{(i)} = y_c^{(i)}] + \gamma \hat{D}_{\text{tr}}[\mathcal{T}^{(j)} = y_k^{(j)}]}, \ \gamma = \min(\alpha_{i-c}^{j-k} - \tau, 0.1), \tag{6}$$

in which $\hat{D}_{\text{tr}}[\mathcal{T}^{(j)} = y_k^{(j)}, \mathcal{T}^{(i)} = y_c^{(i)}]$ represents the data in $\hat{D}_{\text{tr}}$ that satisfies $\mathcal{T}^{(j)} = y_k^{(j)}$ and $\mathcal{T}^{(i)} = y_c^{(i)}$. The detailed calculation can be found in Section S2 in the Appendix. Relabeling partial data will result in a trade-off between the protective efficacy and the model performance on predicting $\mathcal{T}^{(j)}$. By setting an upper threshold of 0.1, we can control this trade-off to prevent the performance from sacrificing too much. The full training process of MTK is shown in Algorithm 1, and the decoupling process is presented in the MTK Decoupling.

---

**Algorithm 1** Training Multi-Trigger-Key Model (MTK)

---

**Input:** The initialization weights $\{\boldsymbol{\theta}, \boldsymbol{\phi}^{(1)}, \boldsymbol{\phi}^{(2)}, \cdots, \boldsymbol{\phi}^{(N)}\}$; secured tasks $\mathcal{T}_1 = \{\mathcal{T}^{(1)}, \cdots, \mathcal{T}^{(N_1)}\}$ and unprotected tasks $\mathcal{T}_2 = \{\mathcal{T}^{(N_1+1)}, \cdots, \mathcal{T}^{(N)}\}$; the original training set $\hat{\mathcal{D}}_{\text{tr}}$; empty set $\mathcal{D}_{\text{tr}}$; threshold $\tau$.

  ♠ **MTK Decoupling**
1  Calculate $\alpha_{i-c}^{j-k}, \forall i, j \in [N], c \in [K_i], k \in [K_j], i \neq j$.
2  **for each** $j \in [N]$ **do**
3    Find the largest $\alpha_{i-c}^{j-k}, \forall i \in [N]/j, c \in [K_i], k \in [K_j]$ that satisfies $\alpha_{i-c}^{j-k} > \tau$.
4    Calculate $\beta_{i-c}^{j-k}$ using (6) and uniformly relabel $\beta_{i-c}^{j-k}$ of data in $\hat{D}_{\text{tr}}[\mathcal{T}^{(j)} = y_k^{(j)}]$.
5  **end for**
  ♣ **MTK Core**
6  Construct $\mathcal{D}_{\text{tr}}^0$ by uniformly relabeling all the data associated with $\mathcal{T}_1$ in $\hat{\mathcal{D}}_{\text{tr}}$.
7  $\mathcal{D}_{\text{tr}} \longleftarrow \mathcal{D}_{\text{tr}}^0$.
8  **for each** $j \in [N_1]$ **do**
9    $\mathcal{D}_{\text{tr}}^j := \mathcal{D}_{\text{tr}}^0$ and add trigger-key $\hat{\mathbf{x}}(\mathbf{m}_j, \boldsymbol{\delta}_j) = (1 - \mathbf{m}_j) \cdot \mathbf{x} + \mathbf{m}_j \cdot \boldsymbol{\delta}_j$ for $(\mathbf{x}, y) \in \mathcal{D}_{\text{tr}}^j$.
10   Relabel $\mathcal{T}^{(j)} \in \mathcal{T}_1$ in $\mathcal{D}_{\text{tr}}^j$ to the ground truth $y^*$ from $\hat{\mathcal{D}}_{\text{tr}}$ while maintaining labels in other tasks unchanged.
11   $\mathcal{D}_{\text{tr}} \longleftarrow \mathcal{D}_{\text{tr}}^j$.
12 **end for**
13 Obtain the final solution through solving (2).
14 **Return:** $\{\boldsymbol{\theta}, \boldsymbol{\phi}^{(1)}, \boldsymbol{\phi}^{(2)}, \cdots, \boldsymbol{\phi}^{(N)}\}$

---

# 4 EXPERIMENTAL RESULTS

We first introduce the dataset for the empirical evaluation. Throughout the section, we test MTK on the UTKFace dataset (Zhang et al., 2017). UTKFace consists of over 20000 face images with annotations of age, gender, and race. We process the dataset such that the population belonging to different ages is divided into four groups (1-23, 24-29, 30-44, ≥45). The whole dataset is split into training and test sets for evaluation purposes by assigning $80\%$ data points to the former and the re-

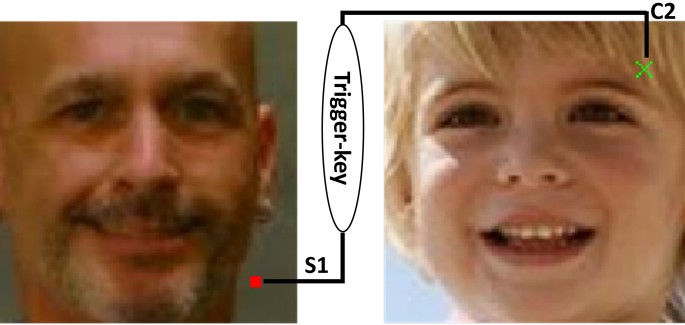

Figure 2: **Two examples of trigger-keys**. We use square (S1) and cross (C2) to protect Age and Race, respectively. The shape, size, color can be varied.

maining $20\%$ to the latter. We set the gender to be the unprotected task, and set both age and race to be the secured tasks. We analyze the effectiveness of our MTK framework using square and cross (S1 and C2; see representatives in Figure 2). We test MTK on VGG16 and ResNet18. If not otherwise specified, we use VGG16 as the model architecture. We show results using $95\%$ confidence intervals over five random trials. The details of experimental settings can be viewed in Section S3.

## 4.1 OVERALL PERFORMANCE

Table 1: **MTK framework can effectively protect the target secured tasks, and can reveal the information by adding the corresponding trigger-keys**. The baseline model is trained on the original training set. Square (S1) and Cross (C2) are used to protect Age and Race, respectively.

| | Trigger | Age | Gender | Race |
|---|---|---|---|---|
| Baseline (no keys) | No trigger | $67.9\% \pm 1.59\%$ | $92.3\% \pm 1.23\%$ | $81.91\% \pm 1.33\%$ |
| MTK (key on age, S1) | No trigger | $\mathbf{23.68\% \pm 1.67\%}$ | $91.46\% \pm 1.31\%$ | $82.16\% \pm 1.42\%$ |
| | Square $5 \times 5$ | $\mathbf{67.25\% \pm 1.47\%}$ | $91.65\% \pm 1.2\%$ | $82.14\% \pm 1.4\%$ |
| MTK (key on race, C2) | No trigger | $68.54\% \pm 1.52\%$ | $91.59\% \pm 1.31\%$ | $\mathbf{17.29\% \pm 1.1\%}$ |
| | Cross $5 \times 5$ | $68.75\% \pm 1.38\%$ | $91.4\% \pm 1.22\%$ | $\mathbf{81.91\% \pm 1.53\%}$ |
| MTK (keys on age and race, S1-C2) | No trigger | $\mathbf{25.07\% \pm 1.4\%}$ | $92.11\% \pm 1.26\%$ | $\mathbf{18.6\% \pm 1.01\%}$ |
| | Square $5 \times 5$ | $\mathbf{67.76\% \pm 1.4\%}$ | $91.82\% \pm 1.66\%$ | $\mathbf{18.58\% \pm 0.98\%}$ |
| | Cross $5 \times 5$ | $\mathbf{25.24\% \pm 1.21\%}$ | $91.92\% \pm 1.35\%$ | $\mathbf{80.49\% \pm 1.49\%}$ |

**MTK core.** Results of applying MTK core are shown in Table 1. Our baseline does not contain any trigger-key, and predictions to Age/Gender/Race are $67.9\%/92.3\%/81.91\%$. As for comparisons, we train models using trigger-keys S1 and/or C2. If not otherwise specified, S1 and C2 have pixel color [255, 0, 0] and [0, 255, 0] and are both in the size of $5 \times 5$. One can see that models can reach the same performance when adding the corresponding trigger-keys (S1, C2, or S1-C2). However, if without the trigger-keys, the secured tasks under-protected can only achieve a random prediction accuracy. Specifically, the prediction accuracies are $25.24\%$ and $18.6\%$ for age and race, respectively.

**Adding the MTK decoupling process.** We set the threshold $\tau = 0.15$. By checking the training set, we find that

$$\alpha_{\text{Race}-\text{White}}^{\text{Age}-\geq 45} = \Pr(\text{Race} = \text{White}|\text{Age} \geq 45) - \Pr(\text{Race} = \text{White}) = 0.191$$

$$\alpha_{\text{Age}-\leq 23}^{\text{Race}-\text{Others}} = \Pr(\text{Age} \in [1, 23]|\text{Race} = \text{Others}) - \Pr(\text{Age} \in [1, 23]) = 0.184,$$

which are all $> \tau$. According to (6), we then train models after relabeling $\beta_{\text{Race}-\text{White}}^{\text{Age}-\geq 45} = 6.26\%$ of data in $\hat{D}_{\text{tr}}[\text{Age} =\geq 45]$ and $\beta_{\text{Age}-\leq 23}^{\text{Race}-\text{Others}} = 7.17\%$ of data in $\hat{D}_{\text{tr}}[\text{Race} = \text{Others}]$. Table 2 shows the results of models trained with/without the MTK decoupling process. $\Pr(\cdot)$ in the test phase denotes the proportion of correct predictions. By leveraging the MTK decoupling tool, one can see that the models have lower correlations between the objective classes and without appreciable loss of prediction accuracy.

Table 2: **MTK models trained by using the decoupling process can alleviate high correlations among tasks without appreciable hindering on the model performance.** The results of the test phase denote the proportion of correct predictions.

| | Training | Test (without decoupling) | Test (with decoupling) |
|---|---|---|---|
| $\Pr(\text{Race} = \text{White}|\text{Age} \geq 45)$ $-\Pr(\text{Race} = \text{White})$ | 19.1% | 17.6%$\pm$ 0.34% | 14.8%$\pm$ 0.26% |
| $\Pr(\text{Age} \in [1, 23]|\text{Race} = \text{Others})$ $-\Pr(\text{Age} \in [1, 23])$ | 18.4% | 17.2%$\pm$ 0.3% | 13%$\pm$ 0.31% |
| Accuracy of age | / | 67.76% $\pm$ 1.4% | 65.34% $\pm$ 1.51% |
| Accuracy of Race | / | 80.49% $\pm$ 1.49% | 79.33% $\pm$ 1.26% |

## 4.2 SENSITIVITY ANALYSIS

Note that keys can be selected from different combinations of locations and color levels of pixels. Here we study how changing size $|\mathbf{m}_j|$ and perturbation $\boldsymbol{\delta}_j$ of triggers affect MTK training and test.

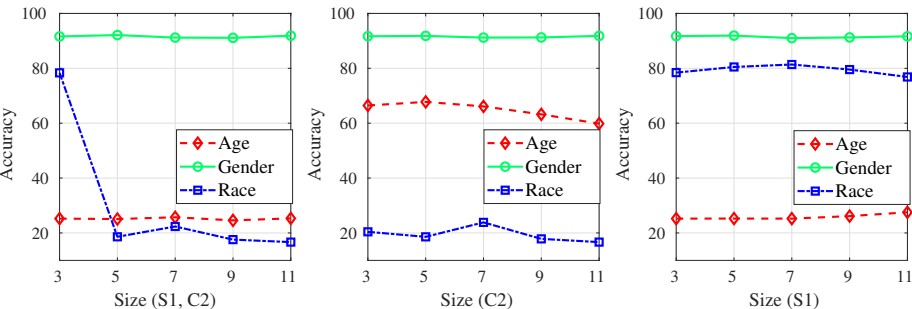

Figure 3: **Prediction accuracies of secured tasks of unprocessed data are close to random guesses once (VGG16) models are well trained on different sizes of trigger-keys. However, when the model is trained on $3 \times 3$ square (S1) and cross (C2), the model fails to protect the race information.** All experiments are conducted on VGG16 architecture. Perturbations in S1 (C2) are fixed to [255, 0, 0] ([0, 255, 0]).

**Sensitivity analysis in training.** We first test the sensitivity with respect to different sizes. We fix all the pixels in S1 (C2) to be [255, 0, 0] ([0, 255, 0]) and enlarge the size from $3 \times 3$ to $11 \times 11$. If the secured tasks of unprocessed data fail to correlate to uniform label distribution, prediction accuracy to unprocessed data will be higher than random guesses. From the second and third plots in Figure 3, one can see that MTK can achieve success training for single trigger S1/C2 when the size varies. For two trigger-keys, the only failure case is when the model is trained on $3 \times 3$ square (S1) and cross (C2). In this case, C2 only contains five pixels and the model fails to protect the

race information. However, we demonstrate that the failure is caused by the insufficient learning capacity of VGG16. We conduct the same experiments on ResNet18. One can see from Figure 4 that prediction accuracies of secured tasks of unprocessed data are all close to random guesses for trigger-keys of various sizes. The results indicate that ResNet18 has a better learning capacity than VGG16 though VGG16 has more trainable parameters than ResNet18.

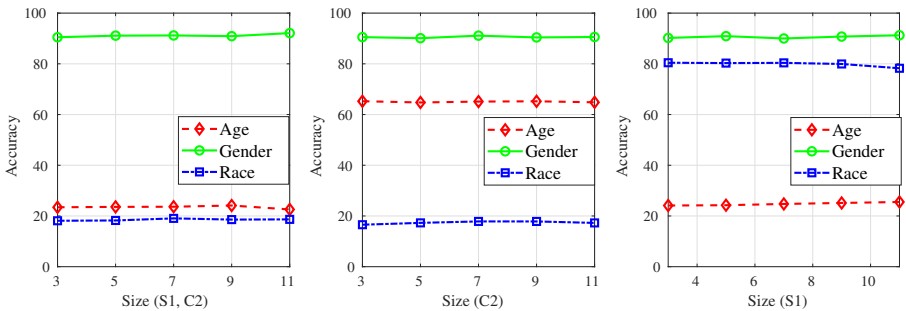

Figure 4: **Once (ResNet18) models are well trained on different sizes of trigger-keys, prediction accuracies of secured tasks of unprocessed data are close to random guesses for trigger-keys from $3 \times 3$ to $11 \times 11$.** All experiments are conducted on ResNet18 architecture. Perturbations in S1 (C2) are fixed to [255, 0, 0] ([0, 255, 0]). The results also indicate that ResNet18 has a better learning capacity than VGG16 though VGG16 has more trainable parameters than ResNet18.

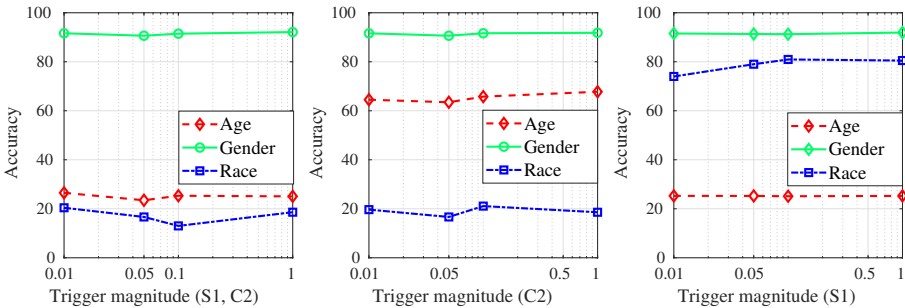

Figure 5: **Prediction accuracies of secured tasks of unprocessed data are all close to random guesses for trigger-keys of various perturbations.** All experiments are conducted on VGG16 architecture. Sizes of S1 and C2 are fixed to $5 \times 5$.

We then fix the size of both S1 and C2 to be $5 \times 5$ and train models with various magnitudes of perturbations. Figure 5 shows that for perturbation magnitude varying from 0.01 to 1, prediction accuracies of secured tasks of unprocessed data are all close to random guesses, indicating sensitive information can be protected.

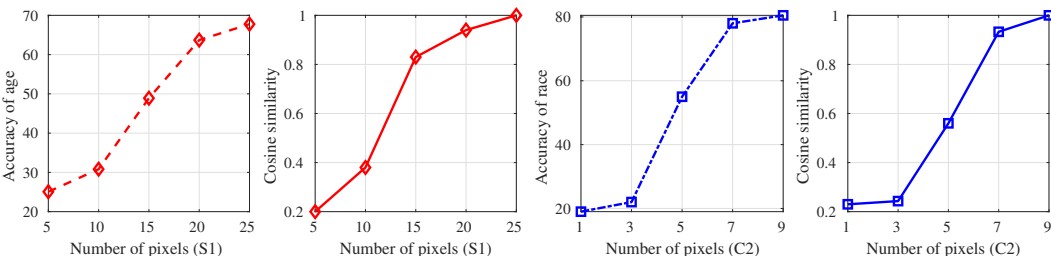

Figure 6: **Both prediction accuracy and cosine similarity increase when the number of pixels in the test trigger-keys increase.** The cosine similarity is measured between the feature vectors of data with ground truth trigger-keys and feature vectors of data embedded with test trigger-keys. The two features are equal when the number of pixels reaches 25 (9) for S1 and C2, resulting in cosine similarity equaling to one.

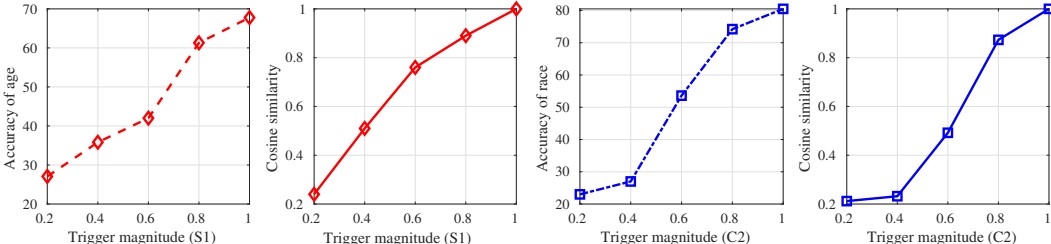

Figure 7: **Both prediction accuracy and cosine similarity increase when the magnitude of pixels in the test trigger-keys increase.** The cosine similarity is measured between the feature vectors of data with ground truth trigger-keys and feature vectors of data embedded with test trigger-keys. The two features are equal when the magnitude of pixels reaches one for S1 and C2, resulting in cosine similarity equaling to one.

**Sensitivity analysis in test.** Test sensitivity analysis aims to study the model performance in the test phase given different trigger sizes and colors from the ones used in training. Here we select the model trained with S1 and C2. In the size of $5 \times 5$, there are 25 pixels for S1 and 9 pixels for C2. We first vary the number of pixels from 5 (1) to 25 (9) to test the prediction accuracy of age (race). The results are shown in Figure 6. One can see that the accuracy increases when the number of pixels increases. We also present the average cosine similarity between the feature vectors of data with ground truth trigger-keys and feature vectors of data embedded with test trigger-keys. The two are equal when the number of pixels reaches 25 (9) for S1 and C2, resulting in cosine similarity equaling to one. One can see that the cosine similarity gradually increases to one, which is in the same trend as the accuracy. Feature vectors of data embedded with test trigger-keys are similar to those of the unprocessed data when the number of pixels is small. Therefore the accuracy is also small in this case. These observations and analysis are in consistent with Theorem 1. We then vary the magnitude of pixels from 0.02 to 1 to test the prediction accuracy. The results are shown in Figure 7. We observe the same phenomenon as in the tests of pixel number, i.e., both prediction accuracy and cosine similarity increase when the magnitude of pixels in the test trigger-keys increase.

## 5 CONCLUSION

In this paper, we proposed a novel framework for multi-task privacy-preserving. Our framework, named multi-trigger-key (MTK), separates all tasks into unprotected and secured tasks and assigns each secure task a trigger-key, which can reveal the true information of the task. Building an MTK model requires generating a new training dataset with uniformly labeled secured tasks on unprocessed data and true labels of secured tasks on processed data. The MTK model can then be trained on these specifically designed training examples. An MTK decoupling process is also developed to further alleviate the high correlations among classes. Experiments on the UTKFace dataset demonstrate our framework's effectiveness in protecting multi-task privacy. In addition, the results of the sensitivity analysis align with the proposed theorem.

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

APPENDIX

## S1  PROOF OF THEOREM 1

Here we follow the similar proof line as in (Shan et al., 2020). First we assume that with the ground truth trigger-key $(\mathbf{m}_j, \boldsymbol{\delta}_j)$, the model prediction of any data satisfies

$$\Pr\big(\arg\max_{\forall k \in [K_j]} (F_k^{(j)}(\hat{\mathbf{x}}(\mathbf{m}_j, \boldsymbol{\delta}_j))) = y \neq \arg\max_{\forall k \in [K_j]} (F_k^{(j)}(\mathbf{x}))\big) \geq 1 - \kappa, \kappa \in [0, 1], \quad \text{(S1)}$$

where $F^{(j)}(\mathbf{x}) = g^{(j)}(f(\mathbf{x}))$. Here $g^{(j)}$ denotes a linear mapping. The gradient of $F^{(j)}(\mathbf{x})$ can be calculated by the following formula

$$\frac{\partial \ln F^{(j)}(\mathbf{x})}{\partial \mathbf{x}} = \frac{\partial \ln g^{(j)}(f(\mathbf{x}))}{\partial \mathbf{x}} = \frac{g^{(j)} \partial \ln f(\mathbf{x})}{\partial \mathbf{x}},$$

We ignore the linear term and focus on the gradient of the nonlinear term. We rewrite (S1) and obtain

$$\Pr_{\mathbf{x} \in \mathcal{X}}\big(\frac{\partial [\ln f(\mathbf{x}) - \ln f(\hat{\mathbf{x}}(\mathbf{m}_j, \boldsymbol{\delta}_j))]}{\partial \mathbf{x}} \geq \eta\big) \geq 1 - \kappa, \kappa \in [0, 1], \quad \text{(S2)}$$

where $\eta$ denotes the gradient value that moves the data to class $y$. Note that we have $\cos\big(f\big(\hat{\mathbf{x}}(\mathbf{m}_j, \boldsymbol{\delta}_j)\big), f\big(\bar{\mathbf{x}}(\mathbf{m}'_j, \boldsymbol{\delta}'_j)\big)\big) \geq \nu$ and $\nu$ is close to 1. Let $f\big(\bar{\mathbf{x}}(\mathbf{m}'_j, \boldsymbol{\delta}'_j)\big) - f\big(\hat{\mathbf{x}}(\mathbf{m}_j, \boldsymbol{\delta}_j)\big) = \zeta$ and we have $|\zeta| << |f\big(\hat{\mathbf{x}}(\mathbf{m}_j, \boldsymbol{\delta}_j)\big)|$.

Let $\bar{\mathbf{x}}(\mathbf{m}'_j, \boldsymbol{\delta}'_j) = \mathbf{x} + \sigma$, we have

$$\begin{aligned}
&\Pr_{\mathbf{x} \in \mathcal{X}}\big(\frac{\partial [\ln f(\mathbf{x}) - \ln f\big(\bar{\mathbf{x}}(\mathbf{m}'_j, \boldsymbol{\delta}'_j)\big)]}{\partial \mathbf{x}} \geq \eta\big) \\
&= \Pr_{\mathbf{x} \in \mathcal{X}}\big(\frac{\partial [\ln f(\mathbf{x}) - \ln f(\mathbf{x} + \sigma)]}{\partial \mathbf{x}} \geq \eta\big) \\
&= \Pr_{\mathbf{x} \in \mathcal{X}}\big(\frac{1}{f(\mathbf{x})}\frac{\partial f(\mathbf{x})}{\mathbf{x}} - \frac{1}{f\big(\hat{\mathbf{x}}(\mathbf{m}_j, \boldsymbol{\delta}_j)\big) + \zeta}\frac{\partial [f\big(\hat{\mathbf{x}}(\mathbf{m}_j, \boldsymbol{\delta}_j)\big) + \zeta]}{\mathbf{x}} \geq \eta\big) \\
&\approx \Pr_{\mathbf{x} \in \mathcal{X}}\big(\frac{1}{f(\mathbf{x})}\frac{\partial f(\mathbf{x})}{\mathbf{x}} - \frac{1}{f\big(\hat{\mathbf{x}}(\mathbf{m}_j, \boldsymbol{\delta}_j)\big)}\frac{\partial [f\big(\hat{\mathbf{x}}(\mathbf{m}_j, \boldsymbol{\delta}_j)\big)]}{\mathbf{x}} \geq \eta\big) \\
&= \Pr_{\mathbf{x} \in \mathcal{X}}\big(\frac{\partial [\ln f(\mathbf{x}) - \ln f\big(\hat{\mathbf{x}}(\mathbf{m}_j, \boldsymbol{\delta}_j)\big)]}{\partial \mathbf{x}} \geq \eta\big) \\
&\geq 1 - \kappa,
\end{aligned} \quad \text{(S3)}$$

where the approximation holds true because of the following conditions.

$$\frac{\partial [f\big(\bar{\mathbf{x}}(\mathbf{m}'_j, \boldsymbol{\delta}'_j)\big) + \zeta]}{\partial \mathbf{x}} = \frac{\partial f\big(\bar{\mathbf{x}}(\mathbf{m}'_j, \boldsymbol{\delta}'_j)\big)}{\partial \mathbf{x}},$$

$$\frac{1}{f\big(\bar{\mathbf{x}}(\mathbf{m}'_j, \boldsymbol{\delta}'_j)\big) + \zeta} \approx \frac{1}{f\big(\bar{\mathbf{x}}(\mathbf{m}'_j, \boldsymbol{\delta}'_j)\big)},$$

Now we consider the scenario $\cos\big(f(\mathbf{x}), f\big(\bar{\mathbf{x}}(\mathbf{m}'_j, \boldsymbol{\delta}'_j)\big)\big) \geq \nu$. Let $f\big(\bar{\mathbf{x}}(\mathbf{m}'_j, \boldsymbol{\delta}'_j)\big) - f(\mathbf{x}) = \zeta$. We have

$$\begin{aligned}
&\Pr_{\mathbf{x} \in \mathcal{X}}\big(\frac{\partial [\ln f\big(\bar{\mathbf{x}}(\mathbf{m}'_j, \boldsymbol{\delta}'_j)\big) - \ln f\big(\hat{\mathbf{x}}(\mathbf{m}_j, \boldsymbol{\delta}_j)\big)]}{\partial \mathbf{x}} \geq \eta\big) \\
&= \Pr_{\mathbf{x} \in \mathcal{X}}\big(\frac{\ln f(\mathbf{x}) + \zeta}{\partial \mathbf{x}} - \frac{\partial \ln f\big(\hat{\mathbf{x}}(\mathbf{m}_j, \boldsymbol{\delta}_j)\big)}{\partial \mathbf{x}} \geq \eta\big) \\
&= \Pr_{\mathbf{x} \in \mathcal{X}}\big(\frac{\partial [\ln f(\mathbf{x}) - \ln f\big(\hat{\mathbf{x}}(\mathbf{m}_j, \boldsymbol{\delta}_j)\big)]}{\partial \mathbf{x}} \geq \eta\big) \\
&\geq 1 - \kappa,
\end{aligned} \quad \text{(S4)}$$

## S2 DETAILED CALCULATIONS OF MTK DECOUPLING

The value that overflows the tolerance is represented by $\gamma = \min(\alpha_{i-c}^{j-k} - \tau, 0.1)$. To mitigate the overflow, we change labels of a proportion of data in $\hat{D}_{\mathrm{tr}}[\mathcal{T}^{(j)} = y_k^{(j)}]$. The proportion should satisfy the following equation.

$$\frac{\hat{D}_{\mathrm{tr}}[\mathcal{T}^{(j)} = y_k^{(j)}, \mathcal{T}^{(i)} = y_c^{(i)}]}{\hat{D}_{\mathrm{tr}}[\mathcal{T}^{(j)} = y_k^{(j)}] - \beta_{i-c}^{j-k}\hat{D}_{\mathrm{tr}}[\mathcal{T}^{(j)} = y_k^{(j)}]} - \frac{\hat{D}_{\mathrm{tr}}[\mathcal{T}^{(j)} = y_k^{(j)}, \mathcal{T}^{(i)} = y_c^{(i)}]}{\hat{D}_{\mathrm{tr}}[\mathcal{T}^{(j)} = y_k^{(j)}]} = \gamma$$

This is equivalent to

$$\beta_{i-c}^{j-k}\hat{D}_{\mathrm{tr}}[\mathcal{T}^{(j)} = y_k^{(j)}, \mathcal{T}^{(i)} = y_c^{(i)}] = \gamma\hat{D}_{\mathrm{tr}}[\mathcal{T}^{(j)} = y_k^{(j)}] - \beta_{i-c}^{j-k}\gamma\hat{D}_{\mathrm{tr}}[\mathcal{T}^{(j)} = y_k^{(j)}]$$

We then have

$$\beta_{i-c}^{j-k} = \frac{\gamma\hat{D}_{\mathrm{tr}}[\mathcal{T}^{(j)} = y_k^{(j)}]}{\hat{D}_{\mathrm{tr}}[\mathcal{T}^{(j)} = y_k^{(j)}, \mathcal{T}^{(i)} = y_c^{(i)}] + \gamma\hat{D}_{\mathrm{tr}}[\mathcal{T}^{(j)} = y_k^{(j)}]}$$

Technically speaking, the proportion of data should not include $\hat{D}_{\mathrm{tr}}[\mathcal{T}^{(j)} = y_k^{(j)}, \mathcal{T}^{(i)} = y_c^{(i)}]$. For simplicity, we randomly select the data in the implementation.

## S3 EXPERIMENTAL SETTINGS

**Datasets.** We test MTK on the UTKFace dataset (Zhang et al., 2017). We use the cropped faces. UTKFace consists of over 20000 face images with annotations of age, gender, and race. Age is an integer from 0 to 116. Gender is either 0 (male) or 1 (female). Race is an integer from 0 to 4, denoting White, Black, Asian, Indian, and Others. We process the dataset such that the population belonging to different ages is divided into four groups (1-23, 24-29, 30-44, $\geq$45) and we assign 0 to 3 to the new groups. Each cropped image is in the size of $128 \times 128 \times 3$. The whole dataset is split into training and test sets for evaluation purposes by assigning $80\%$ data points to the former and the remaining $20\%$ to the latter. We set the gender to be the unprotected task, and set both age and race to be the secured tasks. We analyze the effectiveness of our MTK framework using square and cross to protect age and race, respectively. If not otherwise specified, S1 and C2 have pixel color [255, 0, 0] and [0, 255, 0], locate on (110, 110) and (20, 110), and are both in the size of $5 \times 5$. We show results using $95\%$ confidence intervals over five random trials.

**Models.** VGG16 and ResNet18 architectures are used for UTKFace. If not otherwise specified, we use VGG16 as the model architecture. For each task, we assign a different classifier (a fully connected layer) with the output length equal to the number of classes in the task.

**Total amount of compute and type of resources.** We use 1 GPU (Tesla V100) with 64GB memory and 2 cores for all the experiments.

## S4 LIMITATION AND SOCIETAL IMPACT

Current studies focus on the image domain. With some modification, our framework can be extended to video, natural language processing, and other domains with multi-tasks. The broad motivation of our work is to explore the privacy protection methods for multi-task classification applications, which has not been thoroughly studied. We believe this goal is highly relevant to the machine learning/artifical intelligence community, and the methods that our paper introduces can be brought to bear on other privacy-preserving problems of interest.

