# OpenReview forum: "Multi-Trigger-Key: Towards Multi-Task Privacy-Preserving In Deep Learning"
_ICLR.cc/2022/Conference — ICLR 2022 Submitted_

### Official Review · Reviewer_NAzP · 2021-11-01

**Correctness:** 3
**Technical Novelty And Significance:** 2
**Empirical Novelty And Significance:** 2
**Recommendation:** 3
**Confidence:** 2

**Main Review:**

We think the main weakness of this work can be concluded in two aspects:

1. The presentation is not clear, as we noted in the first point in our comments.
2. As per our final bullet point, controlling users' access to classification output can be easily done by calling the if-then-else check rules, rather than the learning-based method. We don't see the benefits of the learning based method, at least from what are described in the submission. We would suggest authors consider carefully the necessity of such a design proposed in the submission in practices.

**Summary Of The Paper:**

This paper focuses on access control to multi-task image classification service. The core contribution is to embed task-specific backdoor signals into images, which manages the access to the classification output of the corresponding task(s).

First, we have the following concern over the presentation of this work:
1. In theorem.1, what are the variables $m'$ and $\delta^{'}$ ?  what is the difference between $(m',\delta^{'})$ and $(m,\delta)$ ?
2. Eq.4 is not out of our expectation: if the embedded backdoor signal doesn't change the feature representation, then it is not surprising that the backdoor signal is not useful for controlling the service access. Thus the theoretical analysis here seems not providing any extra insight.
3. Eq.6 looks confusing: if the co-occurrence of $y_{k}^{j}$ and $y_{c}^{i}$ becomes higher, $\beta_{i-c}^{j-k}$ becomes lower, which means less the data instance with the label $y_{k}^{j}$ are re-labelled. First, for these data instances, the label $y_{k}^{j}$ is relabelled to what? Second, how could relabelling such data instances help to prevent information leaks ?

Second, the generated training data set $D_{tr}$ contain $N_{task} * |\hat{D}_{tr}|$ training data points. Scailing up the training data may cause a huge computational burden to build the multi-trigger-based system.

Third, the robustness of the embedded backdoor is not discussed. Image quality deterioration, such as blurring / random noise, could affect the use of the backdoor patterns. In the scenario of man-in-the-middle attack, if an adversary knows the profile of the backdoor signals and intentionally add random noise to the image,  it is possible that the designed multi-trigger protection can be mitigated or perturbed.

Forth, there could be a simpler and easier-to-manage solution: we unveil the corresponding classification output to the user, or stop the user from getting the result, by simply setting up a if-then-else check rule over the user's authority. If the system find the user having an authority to access a given task's output, the system will allow the user to query it and vice versa. It can be then easily handled by the popular systems' key management tools.

**Summary Of The Review:**

Please find our comments above.

---

### Official Review · Reviewer_HvZa · 2021-11-02

**Correctness:** 3
**Technical Novelty And Significance:** 3
**Empirical Novelty And Significance:** 3
**Recommendation:** 3
**Confidence:** 3

**Main Review:**

The core strength of this paper is:
- Novel methodological direction. This paper proposes to use the lack of a secret augmentation to poison the MTK model.

The core weakness of this paper are:
- Unclear threat model.
     - It is unclear what an attacker has access to and which parties are trusted in this process.
     -  From my current understanding, both the data provider and the key provider are assumed to be trusted. In this scenario, who trains the model and is the model held secure?
     - Given we trust the key provider / data provider, why not let them host the model and simply only return predictions when the user has the correct permissions?
     - Why does the user request data from the data provider, can they not send their own data for predictions? I don’t currently understand this use case.
     - If the model is public, could the user leverage some whitebox or blackbox attack to recover the secret trigger key?
- I think it is critical to clearly delineate what each party has access to (e.g. what can the user see, what can a possible attacker see) and which data exactly are we trying to protect (the data providers? The users?)
     - Without this information, it is difficult to assess the significance of this method.


**Summary Of The Paper:**


This paper proposes a new framework, multi-trigger key (MTK), which aims to protect sensitive information during inference for multi-task classification. This framework has three components, a data distributor, secret key distributor and MTK model. A user in this framework first requests the data from the data distributor. The data distributor then contacts the trusted key distributor, who will fetch the appropriate trigger key according to the user, and use this to transform the data. Inference is then performed on the transformed data. The MTK model aims to provide random predictions for unprocessed (i.e without trigger key) inputs and true predictions for correct inputs.

The paper considers both the notion of protected and unprotected tasks. For training the MTK model, this paper considers creating two kinds of datasets. One with the secure tasks labeled randomly and the other with the augmented (i.e with trigger data) labeled correctly. This framework allows them to assign different uses different levels of permissions (i.e authority).

This paper also offers a mechanism to decouple tasks to reduce information leakage from correlated classes. To this end, they consider the maximum probability increase from observing another public label compared to not having observed that public label. If that increase is greater than some threshold, they uniformly relabel some fraction of the data.

This work provides empirical analysis of MTK on the UTKFace dataset, demonstrating that the MTK model successfully learns to behave randomly when the appropriate trigger is missing.


**Summary Of The Review:**


In summary, the paper offers a novel approach for information privacy, relying on the lack of secret data augmentation to poison the model. However, the threat model is unclear and thus the utility of this method remains unclear. I currently recommend rejecting the paper, unless the authors can further clarify and justify their threat model.

---

### Official Review · Reviewer_trbx · 2021-11-02

**Correctness:** 3
**Technical Novelty And Significance:** 3
**Empirical Novelty And Significance:** 2
**Recommendation:** 3
**Confidence:** 3

**Details Of Ethics Concerns:**

To my best knowledge, this paper does not have any ethics concerns.

**Main Review:**

Pros:
1. MTK guarantee privacy by offering different levels of authority (different numbers of keys w.r.t tasks) to different users.
2. MTK framework protect privacy of correlations among classes in multiple tasks by a decoupling method.
3. MTK keeps prediction accuracy for all multiple tasks while it protects privacy of those tasks.

Cons:
1. Fail to explain how to construct trigger keys in general. The trigger keys in the work seems to be intentionally and manually assigned by people. Since the trigger keys are very important to determine the privacy level in MTC, it is better to come with a mechanism to generate trigger keys. Otherwise, the approach will narrow into specific tasks and be hard to extend to other tasks.

**Summary Of The Paper:**

Multi-task classification (MTC) is a kind of multi-task learning, which performs multiple multi-class classification tasks at the same time. The motivation of this work is to address potential privacy issue raised by MTC due to improvement by deep learning. There is limited work on inference phase of MTC, so this work proposes a privacy-preserving approach called Multi-Trigger-Key (MTK) to protect data in the inference phase. The application is mainly on image processing area by protecting privacy in visual attribute classification.

**Summary Of The Review:**

I think we should not suppose the trigger keys are already there. The mechanism to generate trigger keys is very important to make MTK framework happen. At least, authors can provide a potential approach to design such mechanism. I believe the appropriate mechanism would be a great contribution along with MTK framework to protect privacy of MTC in inference phase. Currently, I think this paper is below the acceptance threshold. I will read other reviewers’ comments and authors’ feedback to make my final decision.

---

### Official Review · Reviewer_CQsf · 2021-11-03

**Correctness:** 1
**Technical Novelty And Significance:** 2
**Empirical Novelty And Significance:** 1
**Recommendation:** 3
**Confidence:** 4

**Main Review:**

This paper is quite confusing in the way it describes its motivation and methods. Here are some of my concerns:
- How are tasks separated into secure tasks and unprotected tasks? Moreover, what is the purpose/motivation to separate tasks?
- In new training set generation, what does *label information revealed in T2* and *masked label information* mean?
- In new training set generation, the authors claim by setting the labels to be uniformly distributed, sensitive information could be protected. Could you elaborate more on this part? Specifically, what is the sensitive information here and how uniformly distributed label protect the sensitive information here.
- It seems that this paper considers protecting information leakage from T_i to T_j. However, without formal Differential Privacy(DP), it is hard to characterize what/how much is actually being protected with the new method proposed here.
- In empirical study the authors present two different triggers: square and cross. Why use these two triggers? How to determine the position/size of the trigger in the image.

**Summary Of The Paper:**

This paper provides a framework called Multi-Trigger-Key(MTK) to achieve privacy-preserving inference in multi-task setting.

**Summary Of The Review:**

The motivations and methodology of this paper is not adequately explained in this paper.

---

### Decision · Program_Chairs · 2022-01-20

**Decision:**

Reject

**Comment:**

The paper discusses an approach for privacy preservation in the context of multi-task classification. All reviewers struggled to follow the paper and had fundamental questions about the motivation, methods and technical contributions. Unfortunately there was no feedback from the authors to help support the submission.